# Aberrantly Expressed RECQL4 Helicase Supports Proliferation and Drug Resistance of Human Glioma Cells and Glioma Stem Cells

**DOI:** 10.3390/cancers12102919

**Published:** 2020-10-11

**Authors:** Sylwia K. Król, Agnieszka Kaczmarczyk, Kamil Wojnicki, Bartosz Wojtas, Bartłomiej Gielniewski, Wieslawa Grajkowska, Katarzyna Kotulska, Cezary Szczylik, Ryszard Czepko, Mariusz Banach, Wojciech Kaspera, Wojciech Szopa, Andrzej Marchel, Tomasz Czernicki, Bozena Kaminska

**Affiliations:** 1Laboratory of Molecular Neurobiology, Nencki Institute of Experimental Biology of the Polish Academy of Sciences, 02-093 Warsaw, Poland; sylwia_krol15@wp.pl (S.K.K.); a.kaczmarczyk@nencki.edu.pl (A.K.); k.wojnicki@nencki.edu.pl (K.W.); b.wojtas@nencki.edu.pl (B.W.); b.gielniewski@nencki.edu.pl (B.G.); 2Depts. Neurology and Neuropathology, Children’s Memorial Health Institute, 04-730 Warsaw, Poland; W.Grajkowska@ipczd.pl (W.G.); K.Kotulska@ipczd.pl (K.K.); 3Department of Oncology, Military Institute of Medicine, 04-141 Warsaw, Poland; cszczylik@wp.pl; 4Department of Neurosurgery, Scanmed S.A. St. Raphael Hospital, Andrzej Frycz Modrzewski Krakow University, 31-705 Krakow, Poland; ryszard.czepko@scanmed.pl (R.C.); mariusz.banach@scanmed.pl (M.B.); 5Department of Neurosurgery, Medical University of Silesia, Regional Hospital, 04-730 Sosnowiec, Poland; wkaspera@sum.edu.pl (W.K.); wszopa@sum.edu.pl (W.S.); 6Department of Neurosurgery, Medical University of Warsaw, 02-097 Warsaw, Poland; andrzej.marchel@wum.edu.pl (A.M.); tomasz.czernicki@wum.edu.pl (T.C.)

**Keywords:** RecQ helicases, gliomas, gene knockdown, proliferation, drug sensitivity

## Abstract

**Simple Summary:**

Human RecQ helicases participate in DNA replication, repair and transcription. Due to important roles in many cellular processes, deregulation of these helicases in cancer could be tumour promoting. We found upregulated *RECQL4* expression in highly malignant brain tumours (glioblastomas) associated with poor survival of patients. While RECQL4 depletion in human glioma cells with genetic tools slightly impaired cell viability and DNA replication, it induced gross changes in gene expression and increased sensitivity of glioma cells and glioma stem cells to chemotherapeutics. Therefore, targeting RECQL4 in deadly brain tumours could be a new strategy to improve eradication of tumour cells by chemotherapeutics.

**Abstract:**

Anti-tumour therapies eliminate proliferating tumour cells by induction of DNA damage, but genomic aberrations or transcriptional deregulation may limit responses to therapy. Glioblastoma (GBM) is a malignant brain tumour, which recurs inevitably due to chemo- and radio-resistance. Human RecQ helicases participate in DNA repair, responses to DNA damage and replication stress. We explored if a helicase RECQL4 contributes to gliomagenesis and responses to chemotherapy. We found upregulated *RECQL4* expression in GBMs associated with poor survival of GBM patients. Increased levels of nuclear and cytosolic RECQL4 proteins were detected in GBMs on tissue arrays and in six glioma cell lines. RECQL4 was detected both in cytoplasm and mitochondria by Western blotting and immunofluorescence. RECQL4 depletion in glioma cells with siRNAs and CRISPR/Cas9 did not affect basal cell viability, slightly impaired DNA replication, but induced profound transcriptomic changes and increased chemosensitivity of glioma cells. Sphere cultures originated from RECQL4-depleted cells had reduced sphere forming capacity, stronger responded to temozolomide upregulating cell cycle inhibitors and pro-apoptotic proteins. RECQL4 deficiency affected mitochondrial network and reduced mitochondrial membrane polarization in LN18 glioblastoma cells. We demonstrate that targeting RECQL4 overexpressed in glioblastoma could be a new strategy to sensitize glioma cells to chemotherapeutics.

## 1. Introduction

RecQ DNA helicases are enzymes that unwind DNA in an ATP-dependent and directionally specific manner and have overlapping or complementary roles in different DNA metabolic pathways [1]. These enzymes control DNA replication and recombination, transcription, DNA repair mechanisms and telomere maintenance [2,3]. Mutations in genes coding for BLM, WRN and RECQL4 helicases are associated with recessive autosomal syndromes characterized by chromosomal instability, premature aging and predisposition to cancer [3,4]. Mutations in the region encoding the helicase domain of RECQL4 (exons 8–14) occur in the Rothmund–Thomson syndrome (RTS), and fibroblasts from RTS patients show increased sensitivity to genotoxic agents [5,6], but no cellular phenotype has been associated with its deficiency. Deletion of exons encoding the RECQL4 helicase domain in mice is lethal, and rare surviving mice show premature aging [7]. Mutations in *RECQL4* increase the risk of developing breast cancer [8], and two intronic SNPs in *RECQL4* were associated with outcomes of glioblastoma patients [9]. Expression of *RECQL4* is elevated in certain cancer cells, breast and prostate cancer tissues [10,11,12,13]. Knockdown of RECQL4 with short hairpin (sh) RNA in breast and prostate cancer cells increased spontaneous DNA strand breaks, reduced cell survival in vitro and tumour growth in vivo [12,13].

Glioblastoma (GBM) is the most common, primary brain tumour in adults characterized by intense cell proliferation, diffusive growth and aberrant angiogenesis. Despite intensive treatment with post-surgery radiation and chemotherapeutics, GBMs typically recur in 6 months as even more aggressive tumours due to high resistance and frequent dysfunctions in tumour suppressors, oncogenes or apoptotic pathways [14,15]. The presence of glioma stem cells (GSCs) contributes to tumour recurrence [16,17]. Current GBM chemotherapy with an alkylating agent temozolomide (TMZ) prolongs patient survival by several months, but a median overall survival is 14 months after diagnosis [18]. At least 50% of TMZ-treated patients do not respond to TMZ, primarily due to expression of O^6^-methylguanine methyltransferase (MGMT) and/or dysfunctions of DNA repair pathways in GBM cells [19].

We found upregulation of RECQL4 (at mRNA and protein levels) in malignant gliomas and cell lines. To gain insight into the function of RECQL4 in gliomas, we deleted RECQL4 in human glioblastoma cells and analysed consequences of its deficiency on cell growth, viability, stemness capacity, and cell responses to chemotherapeutics. Knockdown of RECQL4 affected slightly glioma cell proliferation, blocked self-renewal of GCSs, and sensitized certain glioma cells to chemotherapy. Due to the presence of RECQL4 in mitochondria, its knockdown impaired mitochondrial networks and membrane potential. Altogether, we demonstrate that targeting upregulated RECQL4 in malignant gliomas may provide a new strategy for anti-glioma therapy.

## 2. Result

### 2.1. RECQL4 Expression Is Highly Upregulated in Glioblastoma Specimens and Cell Lines

Using transcriptomic data from The Cancer Genome Atlas (TCGA) we assessed *RECQL4* expression in human gliomas and normal tissues, and we found upregulation of *RECQL4* mRNA in glioblastomas (WHO grade IV) (Figure 1A). This finding was corroborated by quantification of *RECQL4* mRNAs in 104 glioma samples and 9 normal brain specimens. The levels of *RECQL4* mRNA were higher in high grade gliomas (HGGs) than in normal brains (Figure 1B). Kaplan–Meier analysis demonstrated that survival of HGG patients is negatively associated with *RECQL4* expression (*p* = 0.02) (Figure 1C).

We analysed RECQL4 protein levels using tissue microarrays containing 208 cases of brain malignancies, adjacent tissues and normal brain tissues (Figure 1D). We found numerous specimens with positive RECQL4 staining more abundant in tumours versus normal brain tissues. A detailed immunohistochemical analysis showed more RECQL4 positive cells with strong nuclear or nucleo-cytoplasmic staining in HGGs (Figure 1E). RECQL4 was also overexpressed (on both mRNA and protein levels) in 4 established glioma cell lines and 2 GBM-derived cultures in comparison to normal human astrocytes (NHA) (Figure 1F–G). These results demonstrate upregulated expression of RECQL4 in malignant gliomas and cultured glioma cells.

### 2.2. RECQL4 Knockdown Differently Affects Cell Viability and Proliferation of Glioma Cells

To study RECQL4 function in human glioma cells, we effectively depleted RECQL4 with two siRNAs in LN18 and U87-MG cells, as confirmed by Western blotting (Figure 2A). Transient knockdown of RECQL4 did not decrease cell viability when compared to siCTRL transfected cells (Figure 2B); however, cell proliferation was reduced by 20% in siRQ4 transfected LN18 and U87-MG cells (Figure 2C), as determined with MTT metabolism and BrdU proliferation tests.

Furthermore, we generated two glioma cell lines with RECQL4 knockout (KO #1,2) using CRISPR/Cas9 genome editing. Effectiveness of genome editing was verified by ultra-deep sequencing. RECQL4 depletion in two independent clones of LN18 and LN229 cells was validated by Western blotting (Figure 2D,G). Viability of LN18 RECQL4 KO glioma cells was not affected, while viability of LN229 KO cells decreased by 30–50% (Figure 2E,H respectively). Cell proliferation was significantly inhibited in two RECQL4 KO cell lines, when compared with parental cells, with more profound effects in LN229 RECQL4 KO cells (Figure 2F,I).

### 2.3. RNA Sequencing Reveals Gross Transcriptomic Changes in RECQL4 Depleted Cells

We compared transcriptomes of RECQL4-KO and parental cells by performing RNA sequencing and differential gene expression (DE) analysis. We focused on genes that change consistently in two cell lines. Volcano plots show a large number of genes significantly up- or downregulated (Figure 3). DE analysis revealed 3056 genes downregulated and 1699 genes upregulated in RECQL4 KO samples (FDR < 0.05). KEGG analysis of downregulated genes showed a category of ribosome as the most downregulated pathway and other pathways such as RNA transport, RNA degradation, protein processing in endoplasmic reticulum, aminoacil-tRNA biosynthesis. This analysis suggests alterations of RNA and protein synthesis and processing (Figure 3A,C). KEGG analysis of upregulated genes showed numerous pathways and processes to be modified, including cell cycle, apoptosis (Figure 3D), focal adhesion (Figure 3E), nucleotide excision repair (Figure 3F), mismatch repair (Figure 3G), DNA replication (Figure 3H), p53 signalling pathway (Figure 3I). Expression of many genes involved in cell division and DNA maintenance was increased in RECQL4 KO cells, including *PARP1/2*, *CHECK1/2*, *MSH6*, *POLD1/2/3*, *CDK1/2/7* and many others (Figure 3D–I).

### 2.4. RECQL4 Supports Maintenance of Glioma Stem Cells and Their Resistance to TMZ

Glioblastoma contains glioma stem cells (GSCs), a subpopulation of poorly differentiated or de-differentiated tumour cells with capacity to self-renew and multi-lineage differentiation. GSCs are more resistant to anti-cancer therapeutics and contribute to tumour recurrence [16]. The analysis of TCGA data showed an association of expression of *RECQL4* and well-known stem cell markers *NESTIN, MSI-1* (Musashi-1), *MYC, PROM-1* (CD133) (Figure 4A). LN18 glioma cells grown as non-adherent cells at a low density under serum-free conditions form spheres and display higher expression of stemness markers (*NANOG, POU5F1, SOX2 and PROM-1*) than adherent LN18 cells [20]. We developed glioma sphere cultures from parental and RECQL4 KO LN18 cells and quantified their capacity to self-renew and respond to TMZ. 

Levels of *RECQL4* mRNA and protein were higher in the LN18 spheres than in the adherent cells (Figure 4B,C). Tumour spheres from control (siCTRL) or RECQL4-depleted cells (siRECQL4) were visualized with light microscopy after 7 days of culture, and spheres of ≥ 100 µm in diameter were counted. RECQL4-depleted cells had a significantly reduced number of spheres in comparison to controls (Figure 4D). Expression of *NANOG*, *SOX2* and *POU5F1* was not reduced in RECQL4-depleted spheres, with the exception of *PROM-1* (Appendix A). A number of spheres was even more reduced in two RECQL4 KO clones when compared to parental (WT) cells (Figure 4E).

We found the increased levels of apoptotic markers, such as cleaved PARP-1 and cleaved Caspase 7 in RECQL4 KO spheres. The level of Aurora kinase B (AURKB), one of the regulators of mitosis, was not affected. Phosphorylated histone H2AX level (a marker of double strand DNA breaks and a component of DNA repair machinery) was upregulated (Figure 4F, densitometry shown in the Appendix A). These results suggest that RECQL4 deficiency reduced their capability to self-renew and eliminated some GSCs.

To study if RECQL4 deficiency may affect resistance of GSCs to chemotherapy, WT and RECQL4 KO LN18 spheres were treated with 500 µM TMZ for 72 hours. Light microscopy and quantification of spheres showed a similarly reduced number of spheres in TMZ-treated parental and RECQL4 KO cultures (Figure 4G). Further investigation of proteins involved in apoptosis, cell proliferation and DNA damage repair showed the increased levels of cleaved PARP-1, cleaved Caspases 3 and 7 in TMZ-treated RECQL4 KO #2 spheres compared to WT cells. AURKB and phospho-H2AX levels were more elevated after TMZ in RECQL4 KO #2 spheres compared to WT cells (Figure 4H, densitometry shown in Appendix A) suggesting increased TMZ toxicity in RECQL4 KO spheres.

### 2.5. RECQL4 Deficiency Affects Functionality and Morphology of Mitochondrial Network in Glioblastoma Cells

RECQL4 is the only RecQ helicase found in the nucleus and mitochondrion [21]. We found RECQL4 both in the cytosolic and mitochondrial fractions of U87-MG, LN18 glioblastoma cells, WG4 and IPIN patient-derived cultures by Western blotting (Figure 5A). To study mitochondrial network morphology, WT and RECQL4 KO glioma cells were stained with MitoTracker, and mitochondrial structure was analysed by confocal microscopy. Morphology of the mitochondrial network was affected in RECQL4 KO in both cell lines, with stronger changes in LN18 cells. Mitochondria in parental cells exhibited an elongated shape and formed tubular network-like structures, whereas in RECQL4 KO cells mitochondria had an irregular, disturbed structure, while the mitochondria network was more fragmented and formed short tubules (Figure 5B).

To assess changes of mitochondrial transmembrane potential in parental and RECQL4 KO glioma cells, we stained these cells with a fluorescent, potential sensitive probe-JC-1 and measured fluorescence by flow cytometry. Cells treated with an uncoupling agent CCCP were used as a positive control for mitochondrial membrane depolarization. Representative histograms show changes in the red/green fluorescence intensity ratio of analysed LN18 and LN229 cells. Percentages of cell population are given in the respective quadrants. Note the considerable reduction of % cells with high potential in RECQL4 KO cells. The right panel shows cells with mitochondria depolarized by treatment with CCCP. Only RECQL4 KO LN18 cells showed a statistically significant loss of mitochondrial transmembrane potential (by 40%) in comparison with WT cells (Figure 5C,D).

### 2.6. RECQL4 Deficiency Sensitises LN18 Glioma Cells to Chemotherapeutics

Next, we explored if RECQL4 deficiency will sensitise cells to chemotherapeutics. LN18 glioma cells with unmethylated *MGMT* and non-functional TP53 are resistant to TMZ [22]. Viability of WT and RECQL4 KO LN18 cells exposed for 72 h to 250 and 500 µM TMZ was not significantly altered (Figure 6A), although some dead cells emerged (Appendix A). TMZ reduced cell proliferation in two RECQL4 KO clones by 40–50% but had weak effect on WT cells (Figure 6B). Western blotting analysis revealed stronger increases of cleaved PARP, cleaved Caspase 3 and 7 levels in TMZ-treated RECQL4-deficient LN18 cells than in WT cells. Phosphorylation of CHK1/2, H2AX, TP53 at Ser15 was increased and AURKB levels were more elevated in RECQL4 KO cells after TMZ treatment (Figure 6C and Appendix A).

RECQL4 interacts and may cooperate in DNA repair with Poly(ADP-ribose) polymerase (PARP), and PARP inhibitors show synthetic lethality in diverse cancer cells with defective DNA repair [23,24,25]. We tested if olaparib, a PARP inhibitor, would modulate behaviour of RECQL4-depleted LN18 glioma cells. Olaparib did not significantly affect cell viability or proliferation (Figure 6D,E; Appendix A). Western blot analysis showed weak increases of cleaved PARP, Caspase 7, phospho-CHK1,2, phospho-Ser15-p53 and AURKB levels after olaparib treatment. All those changes were augmented in RECQL4 KO clone #2 cells in comparison to WT cells (Figure 6F and Appendix A). We evaluated cell cycle distribution in WT and RECQL4 KO LN18 cultures treated with TMZ or olaparib by flow cytometry. Reduced percentages of cells in the G1 phase and increased in the G2/M phase were found in TMZ-treated RECQL4 KO LN18 cultures (Figure 6G). Immunofluorescent staining showed appearance of phosphorylated H2AX 48 h after TMZ mostly in RECQL4 KO cells in comparison to WT cells (Figure 6H). In olaparib-treated cultures, distribution of cells in the cell cycle (Figure 6G) or γH2AX foci in RECQL4 KO cells and controls were similar.

LN229 glioma cells have a methylated *MGMT* gene promoter and a wild type TP53 [26]. TMZ (Figure 7A,B) and Olaparib (Figure 7D,E) did not influence cell viability, but affected proliferation of LN229 cells, similarly affecting WT and RECQL KO cells. Western blotting showed increases of both cell cycle regulators (CHK1/2, AURKB), cleaved PARP and active caspases (Figure 7C; densitometry shown in Appendix A). The appearance of floating cells and increased levels of cleaved caspases indicated cell death in TMZ-treated cultures. In LN229 cells treatment with 10 µM olaparib upregulated CHEK1/2, AURKB and weakly activated PARP and Caspase 7 cleavage (Figure 7F and Appendix A). RECQL4 KO LN229 cells responded to TMZ or olaparib similarly as WT cells.

## 3. Discussion

RecQ helicases participate in DNA repair, telomere maintenance and replicative stress and can reduce the efficacy of cancer chemotherapeutics generating DNA damage [27]. Glioblastoma is a chemo-/radio-resistant tumour and recurs in >90% of patients. Drug resistance is attributed to the presence of glioma stem cells and high DNA repair capabilities [28]. Herein, we present new insights regarding contribution of RECQL4 helicase to glioma pathobiology. We found upregulation of RECQL4 expression at mRNA and protein levels in high grade gliomas. High *RECQL4* expression was negatively associated with GBM patient survival. We demonstrate that the helicase localises in nuclei and cytoplasm in high grade gliomas, while in normal brains RECQL4 is mostly restricted to cytoplasm. Expression of RECQL4 was considerably higher in all six human glioma cell lines in comparison to normal astrocytes. Mitochondrial localisation of RECQL4 was verified by Western blotting and immunofluorescence. Interestingly, RECQL4-deficient cells had a disturbed mitochondrial network, and in LN18 cells, it was associated with the reduced mitochondrial potential.

Depletion of RECQL4 in two glioma lines with the highest RECQL4 level and distinct genetic background (with siRNAs or the validated CRISPR/Cas9 construct) did not affect viability of LN18 cells and weakly reduced viability of LN229 cells. Basal proliferation was reduced by 20% in both cell lines. This suggests that cell dependency for RECQL4 could be heterogeneous within glioma cell lines and may depend on the status of another oncogenic driver. Those minor changes in cell growth contrasted with gross transcriptomic changes revealed by the RNA-seq analysis. Consistent transcriptomic changes in RECQL4-depleted LN18 and LN229 cells were profound and affected genes involved in transcription, ribosomes, RNA transport. More detailed search for differentially expressed genes revealed upregulation of genes implicated in nucleotide excision and mismatch repair, DNA replication, cell senescence, inflammation and TP53 signalling pathways along with upregulation of apoptosis and focal adhesion related genes. Upregulation of genes involved in DNA replication and DNA repair could be a compensatory response, while upregulation of cell senescence, inflammation and TP53 signalling-related genes is indicative of cell senescence. Mutations in *RECQL4* in fibroblasts from the Rothmund–Thomson syndrome led to premature senescence of fibroblasts and increased sensitivity to genotoxic agents [5,6].

Glioblastomas are rich glioma stem cells that contribute to tumour recurrence. In vitro this phenomenon can be mimicked by culturing cells under specific conditions and LN18 cells growing as spheres in a serum-free medium are enriched in glioma stem cells [20]. *RECQL4* mRNA was upregulated in LN18 spheres in comparison to adherent cells. RECQL4 depletion reduced a number of spheres suggesting the impaired sphere forming capacity. The augmented γH2AX level indicates an increase in DNA double-strand breaks (DSB), while upregulation of cleaved PARP and cleaved Caspase 7 in RECQL4 KO cells shows ongoing cell death. RECQL4-depleted LN18 cells were more sensitive to TMZ treatment than parental cells, as indicated by stronger accumulation of apoptotic proteins. This evidence points to a new role of RECQL4 in glioma stemness and chemosensitivity.

We explored if RECQL4 depletion in glioma cells could affect their sensitivity to TMZ or a PARP inhibitor. While WT LN18 glioma cells were resistant to TMZ or olaparib, RECQL4 KO cells were more sensitive to TMZ. Increased levels of proteins related to cell cycle, apoptosis and DBS in LN18 RECQL4 KO cells demonstrate their increased sensitivity to TMZ. Upregulation of phospho-CHK1 and phospho-CHK2 in RECQL4 KO cells indicates growth arrest, which was confirmed by a shift in percentages of cells in the G1 phase and the G2/M phase. Interestingly, we found that RECQL4 KO glioma cells overcome TP53 deficiency after TMZ treatment TMZ and activate growth arrest and apoptotic pathways. Interestingly, RECQL4-depleted LN229 glioma cells with functional TP53 and reduced MGMT expression responded to chemotherapeutics as well as WT cells.

Inhibition of PARP is toxic in cells with defective DNA base excision repair and PAPR inhibitors are recently explored as anti-glioma treatments [29,30]. In this study, olaparib had weaker effects on cell cycle regulators, pro-apoptotic proteins, phospho-H2AX levels and did not significantly affect proliferation and viability of LN18 cells. A lack of the additive effect of olaparib with RECQL4 depletion suggests that both manipulations affect a similar DNA repair pathway. RECQL4 interacts physically with PARP and cooperates in DNA repair [31].

RECQL4 plays the oncogenic role in some cancers. Genomic alterations, high *RECQL4* mRNA or protein expression were associated with aggressive behaviour in breast cancers and poor survival. RECQL4 depletion impaired DNA replication and increased chemosensitivity in cultured breast cancer cells [12]. *RECQL4* mRNA was upregulated in gastric cancers and RECQL4 depletion reduced cell proliferation and induced growth arrest [32]. Our results support a notion of a strong contribution of the RECQL4 helicase to chemoresistance of glioma cells. We demonstrate that targeting of RECQL4 in glioma cells lacking functional TP53 could be a potential strategy to sensitize cells to DNA damage inducing drugs. Therefore, RECQL4 is emerging as a novel molecular target for improving individualized glioma therapy.

## 4. Materials and Methods

### 4.1. Clinical Tumour Samples

Fresh-frozen glioma tissue samples were obtained from The Canadian Brain Tumour Tissue Bank (London Health Sciences Centre, Ontario, Canada), The Children’s Memorial Health Institute in Warsaw, Department of Neurosurgery, Scanmed S.A. St. Raphael Hospital in Cracow Department of Neurosurgery, Medical University of Silesia, Regional Hospital in Sosnowiec Neurosurgery Department and Clinic and Medical University of Warsaw. Collecting patient material has been approved by the Committee of Bioethics of the respective hospitals (Ethics Committee of Medical University of Warsaw – KB/54/2016; Ethics Committee of Children’s Memorial Health Institute – 14/KBE/2012; Ethics Committee of Medical University of Silesia―KNW/0022/KB1/46/I/16; Ethics Committee of District Medical Chamber in Krakow―73/KBL/OIL/2015). Each patient provided a written consent for the use of tumour tissues.

### 4.2. DNA and RNA Isolation

DNA and RNA were extracted by homogenization of frozen glioma samples in TRI Reagent® (Thermo Fisher Scientific, Waltham, MA, USA). After incubation with proteinase K (600 µg/mL), nucleic acids were extracted with a phenol-chloroform-isoamyl alcohol followed by chloroform and precipitated with 5 M NaCl and 2 volume of 100% ethanol. After this, centrifugation pellets were washed with 70% ethanol, dried and re-suspended in water. Quality and quantity of nucleic acids were determined by NanoDrop 2000 (Thermo Fisher Scientific, Waltham, MA, USA).

### 4.3. Adherent and Sphere Cultures, Treatments

Human glioma LN18, LN229, U87-MG and U251 cell lines were from American Type Culture Collection (ATCC, Manassas, VA, USA); GBM patient-derived glioma primary cultures WG4 and IPIN20160420 were generated and cultured as previously described [20] in DMEM/Nutrient Mixture F-12, GlutaMAX™ medium, supplemented with 10% FBS (Gibco Life Technologies, Rockville, MD, USA) and antibiotics. Normal human astrocytes (NHA, Lonza Walkersville, MD, USA) were cultured in a commercial medium as described [33]. Temozolomide (TMZ, Sigma-Aldrich, Munich, Germany) was dissolved in water. Olaparib (OLA, MedChemExpress, Monmouth Junction, NJ, USA) was dissolved in DMSO.

For sphere cultures, LN18 glioma cells were seeded at a low density (1500 cells/cm^2^) on non-adherent plates and cultured in DMEM/F-12 GlutaMAX™ supplemented with 2% B27, 20 ng/mL rhuman bFGF, 20 ng/mL rhuman EGF, 0.0002% heparin and antibiotics (for details see Appendix A). Cells were fed every 3 days by adding 1 mL of the fresh medium.

### 4.4. RNA Sequencing and Analyses

Quality and integrity of RNA was assessed with Agilent 2100 Bioanalyzer using a RNA 6000 NanoKit (Agilent Biotechnologies, Santa Clara, CA, USA). PolyA enriched RNA libraries were prepared using the KAPA Stranded mRNA Sample Preparation Kit (Kapa Biosystems, Wilmington, MA, USA). Transcriptomic data were analysed as follows: fastq files were aligned to hg38 human reference genome with STAR program [34], and reads were counted to genes using feature Counts algorithm SUBREAD package [35]. Gene counts were normalized with the FPKM method, and differential analysis was performed using the NOIseq package [36] with the noiseqbio method. Results obtained from LN18 and LN229 glioma cells were combined in one differential analysis; the effect of cell line was considered as a batch effect and removed by using the ARSyNseq method from the NOIseq package. Genes were considered to be differentially expressed (DE) with FDR corrected *p*-value < 0.05. Kyoto Encyclopedia of Genes and Genomes (KEGG) pathway analyses were performed using R package clusterProfiler [37] to annotate the functions of differentially expressed (DE) mRNAs. 

Data on *RECQL4* mRNA levels in gliomas of WHO II, II, IV grades were extracted from TCGA datasets [38,39]. 

### 4.5. Transient and Stable Knockdown of RECQL4 With siRNA and CRISPR/Cas9

Glioma LN18 and U87-MG cells (4 × 10^5^) were transfected using AMAXA SF and SE Cell Line 4D-Nucleofector^TM^ X Kit S, respectively (Lonza, Cologne, Germany) with 20 pmol of a control siRNA (pool non-targeting siRNAs) and two RECQL4-specific siRNAs (ON-TARGET human siRECQL4) (Thermo Scientific, Lafayette, CO, USA). The culture medium was replaced 24 h after transfection.

RECQL4 knockout cells were generated using the custom designed CRISPRCLEAR™ Transfection ready KO kit for the human *RECQL4* gene (# ASK-7010, Applied Stem Cell, Milpitas, CA, USA), including the validated gRNA targeting the *RECQL4* exon 3 (CGCACTCTGAAGCTGACCAC) in the expression vector and a Cas9-Puro plasmid (co-expressing hSpCas9 and puromycin resistance genes). Glioma cells were transfected with both plasmids using Lipofectamine2000 (Invitrogen, Thermo Fisher Scientific, Waltham, MA, USA) for 48 h, followed by puromycin (2 µg/mL) selection for 72 h and incubation in puromycin-free media for additional 48 h. On day 7 post-transfection, the surviving, puro-resistant cells were plated as a single cell to form colonies. After culturing, the clones derived from those single cells were validated for RECQL4 KO by Western Blotting and ultra-deep Next Generation Sequencing (Genomed, Warsaw, Poland). Data from NGS showed deletions of short fragments in the *RECQL4* gene. We marked two independently derived clones, derived from a single puro-resistant cell with the lowest expression of RECQL4 protein (as validated by Western blotting) as KO #1 and KO #2.

### 4.6. Cell Viability, Proliferation and Cell Cycle Analysis

Cell viability was determined using MTT metabolism test, as described [20,33]. Cell proliferation was assessed using ELISA BrdU kit (Roche Diagnostics GmbH, Mannheim, Germany), according to the manufacturer’s protocol. For cell cycle analysis, cells were collected by trypsinization, fixed in 70% ethanol and stained with propidium iodide solution (3.8 mM sodium citrate, 50 mg/ml RNAse A, 500 mg/mL PI, in PBS). DNA content analyses were performed using FACS Calibur flow cytometer (BD Biosciences, Waltham, MA, USA) and the BD CellQuest Pro 6.0 software (BD Biosciences, San Jose, CA, USA). At least 10,000 events were analysed for each sample.

### 4.7. Reverse Transcription and Quantitative PCR

Total RNA was extracted from glioma cells using High Pure RNA Isolation Kit (Roche Diagnostics, Mannheim, Germany) or RNeasy Mini kit (Qiagen, Hilden, Germany) and purified on RNeasy columns. RNA (0.5 µg) was used to synthesize cDNA with oligo(dT)_15_ primers (2.5 µM) and 200 units of SuperScript™ III Reverse Transcriptase (Invitrogen, Thermo Fisher Scientific, Waltham, MA, USA). PCR amplifications were performed in duplicate on cDNA equivalent to 18 ng RNA in 10-µL containing 2× Fast SYBR GREEN PCR Master Mix (Applied Biosystems, Foster City, CA, USA) and primers for *GAPDH* and *RECQL4* (Appendix A) in QuantStudio 12K Flex (Applied Biosystems, Life Technologies, Waltham, MA, USA). Relative quantification of gene expression was determined using the comparative CT method.

### 4.8. Preparation of Protein Extracts and Western Blot Analysis

Whole cell protein lysates were prepared by scraping cells into a lysis buffer containing phosphatase and protease inhibitors as described [33]. Mitochondrial and cytosolic extracts were prepared with Qproteome^®^ Mitochondria Isolation Kit (Qiagen GmbH, Hilden, Germany). Cells (10^7)^ were collected, washed in PBS, lysed for 10 min in Lysis Buffer and centrifuged. Cell pellets were homogenized in Disruption Buffer with a needle (×10), washed and centrifuged. Mitochondrial pellets were re-suspended in a lysis buffer [33]. Immunocomplexes were detected using SuperSignal West Pico^PLUS^ Chemiluminescent Substrate (Thermofisher Scientific, Rockford, IL, USA) and visualized by ChemiDoc Imaging System (Bio-Rad Laboratories, Hercules, CA, USA). HP-conjugated anti-β-Actin or anti-GAPDH antibody were used to control for equal protein loading. Membranes were stripped in 100 mM glycine and 20% SDS buffer, pH 3.0 for 30 min at RT, washed, blocked and re-probed with antibodies (Appendix A). Densitometric analysis was performed using Image Lab ver. 5.2 software (Bio-Rad Laboratories, Hercules, CA, USA).

### 4.9. Immunofluorescence Staining for Mitochondrial Networks

Cells were seeded on coverslips (3 × 10^4^ cells/well) for 30 h. The medium was changed to PBS (with Ca^2+^ and Mg^2+^) to induce starvation conditions and the cells were cultured for 18 h. To visualize mitochondrial network, cells were stained with 250 nM MitoTracker CMXRos (Invitrogen, San Diego, CA, USA) in dark at 37 °C for 20 min, washed 3 times with a complete medium and 3 times with a PBS, fixed with 4% PFA/PBS pH 7.4 for 15 min and permeabilised with 100% methanol for 10 min. Unspecific signals were blocked with 5% normal donkey serum (NDS), 1% bovine serum albumin (BSA) in 0.3% Triton X-100/PBS for 40 min at RT, followed by incubation with anti-RECQL4 Ab (1:200, Novus Biological) in 3% NDS, 1% BSA in 0.3% Triton X-100/PBS at +4 °C overnight. In control experiments, primary antibodies were omitted. Coverslips were washed 3 times with 0.1% BSA, 0.3% Triton X-100/PBS, followed by incubation with secondary antibody conjugated with donkey anti-rabbit Alexa Fluor 488 Ab (1:2000, Invitrogen, Thermo Fisher Scientific, Waltham, MA, USA) for 1 h. After washing with 0.1% BSA/0.3% Triton X-100/PBS, cell nuclei were stained with 1 µg/mL DAPI (Sigma-Aldrich, Munich, Germany) for 10 min at RT in darkness. Coverslips mounted with a fluorescent mounting medium (Dako North America Inc. CA, USA) were visualised using a LSM800 AiryScan confocal microscope. Images were acquired, processed and analysed using ZEN ver. 2.3 software (Carl Zeiss Industrielle Messtechnik GmbH, Oberkochen, Germany).

### 4.10. Evaluation of Mitochondrial Membrane Potential

Mitochondrial membrane potential (ΔΨm) was analysed in WT or RECQL4 KO LN18 and LN229 glioma cells seeded 0.5 × 10^6^ cells/dish. Cells were harvested 48 h after seeding, re-suspended in DMEM and incubated with 5 μM (for LN229 cells) and 10 μM (for LN18 cells) of JC-1 (5,5′,6,6′-tetrachloro-1,1′,3,3′-tetraethylbenzimidazole-carbocyanine iodide, Molecular Probes, Eugene, OR, USA) for 15 min at 37 °C in dark. Stained cells were washed twice in cold PBS, suspended in buffer A (121 mM NaCl, 25 mM HEPES, 5 mM NaHCO_3_, 4.7 mM KCl, 1.2 mM KHPO_4_, 1.2 mM MgSO_4_, 2 mM CaCl_2_, 10 mM glucose, pH 7.4) and analysed using a FACSCalibur with BD CellQuest™ software (BD Biosciences, San Jose, CA, USA). JC-1 stained cells incubated with 10 μM uncoupling agent CCCP (carbonyl cyanide m-chloro-phenylhydrazone; Sigma-Aldrich, Munich, Germany) served a positive control. At least 10,000 events/sample were analysed.

### 4.11. Immunohistochemical Staining of Tissue Microarrays

RECQL4 expression was evaluated on glioma tissue microarrays (US Biomax, Derwood, MD, USA). Paraffin-embedded sections were incubated 10 minutes in 60°C, deparaffinised in xylene, rehydrated in ethanol (100%, 90%, 70%) and washed with water. Epitopes were retrieved by boiling in a pH 6.0 citrate buffer for 35 min. Endogenous peroxidase was blocked in 0.3% H_2_O_2_ in methanol for 30 min followed by blocking with 3% horse serum in 0.2% Triton X-100/PBS. Sections were incubated with anti-RECQL4 Ab (1:200, Novus Biologicals, Centennial, CO, USA) in 3% horse serum, 0.1% Triton X-100/PBS overnight at +4 °C, washed in PBS, incubated with a biotinylated horse anti-rabbit IgG (Vector Laboratories, Burlingame, CA, USA; 1:200), then with horseradish peroxidase-conjugated avidin (ExtrAvidin™−Peroxidase, Sigma-Aldrich, Munich, Germany, 1:200, 0.1% Triton X-100/PBS) for 60 min and with 3,3′-diaminobenzidine. Sections were counterstained with hematoxylin (Sigma-Aldrich, Munich, Germany), washed in PBS, and mounted using a glycerol mounting medium. Images were acquired with the Leica DM4000 B microscope operating with Application Suite ver. 2.8.1 software (Leica Microsystems CMS, Heerbrugg, Switzerland). The scoring was performed by two independent scientist based on positive staining of RECQL4 protein in nucleus, cytoplasm or both. The scoring was marked as 1 if staining was “present” or “0” if absent. The resulting scores were analyzed using chi square test among the glioma grades.

### 4.12. Statistical Analysis

All experiments were performed on 3-4 independent cell passages. Results are expressed as means ± standard error of the mean (SEM). *p*-Values were calculated using two-tailed t test, chi-squared test and one-way or two-way ANOVA followed by appropriate post-hoc test using GraphPad Prism v6 (GraphPad Software, San Diego, CA, USA).

## 5. Conclusions

Altogether, we show elevated expression of RECQL4 in malignant gliomas, which confers survival and proliferative advantage to cancer cells. Reduced sphere forming capacity of sphere cultures originated from RECQL4-depleted cells, and their stronger responses to temozolomide with upregulating cell cycle inhibitors and pro-apoptotic proteins indicate a new role of RECQL4 in stemness maintenance. Increased sensitivity to TMZ (overcoming T53 deficiency), deregulation of mitochondrial network and mitochondrial membrane polarization contribute to greater chemosensitivity of TP53 deficient LN18 glioblastoma cells. All these data combined suggest that targeting RECQL4 overexpressed in glioblastoma could be a new strategy to sensitize glioma cells to chemotherapeutics.

## Figures and Tables

**Figure 1 cancers-12-02919-f001:**
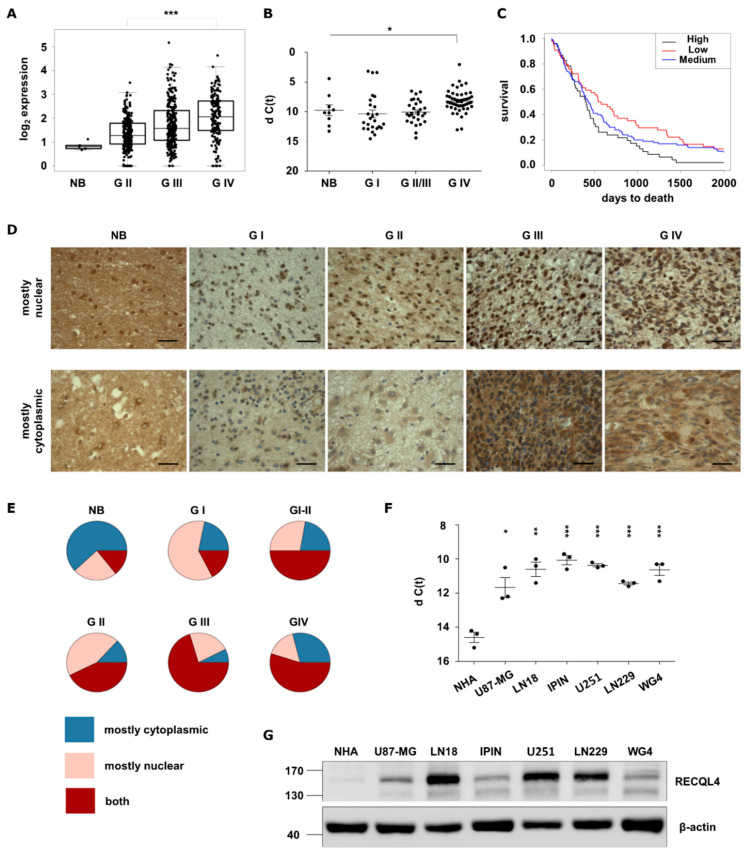
RECQL4 expression is upregulated in human malignant gliomas. (**A**) *RECQL4* expression in normal brain (NB), WHO grade II and grade III gliomas and glioblastomas (GBM, WHO grade IV) in TCGA datasets. Presented values are log_2_ of FPKM values. Statistical significance was determined by Welch’s analysis of variance (ANOVA) between GII, GIII and GIV groups. (**B**) Quantitative analysis of *RECQL4* mRNA levels in NB (*n* = 9), and gliomas of different grades: GI (*n* = 25), GII/III (*n* = 29) and GBM (*n* = 50). The *RECQL4* expression was normalized to *GAPDH*; results represent means ± SEM; statistical significance was determined by one-way ANOVA, followed by Dunnett’s post hoc test. *p*-Values were considered as significant when * *p* < 0.05. (**C**) Kaplan–Meier overall survival analysis of LGG and GBM patients from TCGA. Log-rank test was calculated between *RECQL4* LOW and HIGH expression groups (* *p* < 0.05). (**D**) Representative immunostaining showing expression of RECQL4 protein in the glioma tissue microarray including astrocytomas (*n* = 132), glioblastomas (*n* = 31), oligoastrocytomas (*n* = 7), oligodendrogliomas (*n* = 9), ependymomas (*n* = 11), ganglioglioma (*n* = 1) and gliosarcoma (*n* = 1), plus tumour adjacent and normal brain (NB) tissues (*n* = 8). RECQL4 expression appeared as high nuclear and low cytoplasmic signal (upper panel) and low nuclear and high cytoplasmic signal (bottom panel). (**E**) Quantification of RECQL4 immunoreactivity. Statistical significance was determined by chi-squared test. *p*-values of NB *p* = 1.0, GI *p* = 0.019, GII *p* = 1.1 × 10^−5^, GIII *p* = 6.3 × 10^−5^, GIV *p* = 0.011. (**F**) Quantitative analysis of *RECQL4* mRNA levels in established glioma cell lines, patient-derived primary cultures, and normal human astrocytes (NHA). The expression was normalized to *GAPDH*; results represent means ± SEM of 3 cell passages (*n* =3). Statistical significance was determined by one-way ANOVA. *p*-Values were considered as significant when * *p* < 0.05, ** *p* < 0.01, *** *p* < 0.001. (**G**) Representative immunoblot shows RECQL4 expression in established glioma cell lines and GBM primary cultures in comparison to NHA. β-Actin was used as a loading control.

**Figure 2 cancers-12-02919-f002:**
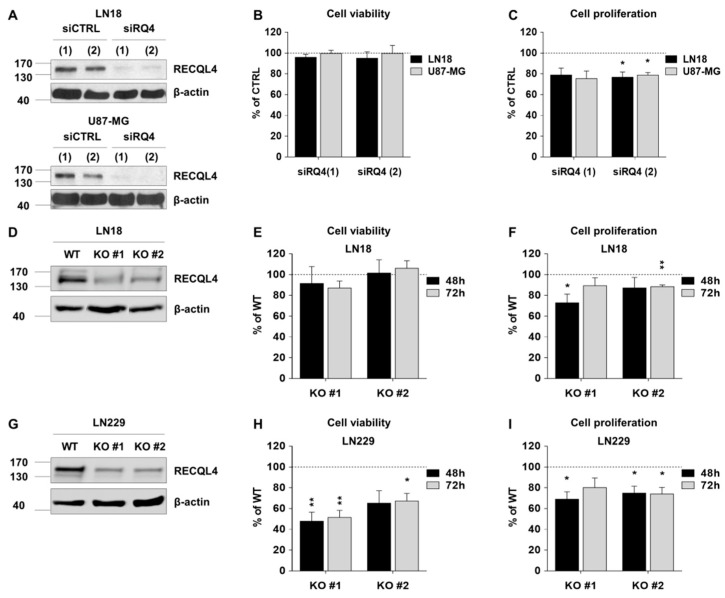
RECQL4 depletion affects cell viability and proliferation of certain glioblastoma cells. (**A**) Representative immunoblots showing effective knockdown of RECQL4 (48 h post-transfection) with two different siRNAs (siRQ) in LN18 (upper panel) and U87-MG (bottom panel) cells. β-Actin was used as a loading control. Viability (**B**) and proliferation (**C**) of LN18 and U87-MG 48 h post-transfection was assessed by MTT metabolism and BrdU incorporation tests, respectively. The results were normalized to cells transfected with the non-targeting control siRNA (siCTRL) and represent the mean ± SEM (*n* = 4). Statistical significance was determined with by two-tailed t-test; statistically significant when * *p* < 0.05. (**D**,**E**) Representative immunoblots show knockout of RECQL4 by CRISPR/Cas9 in two independent clones (RECQL4 KO #1 and KO #2) of LN18 and LN229 cells. Viability (**G**,**H**) and proliferation (**F–I**) of LN18 or LN229 cells were assessed 48 h and 72 h after seeding as described above. The results were normalized to wild type (WT) cells and represent the mean ± SEM of four independent experiments. Statistical significance was determined by one sample t-test; significant when * *p* < 0.05, ** *p* < 0.01.

**Figure 3 cancers-12-02919-f003:**
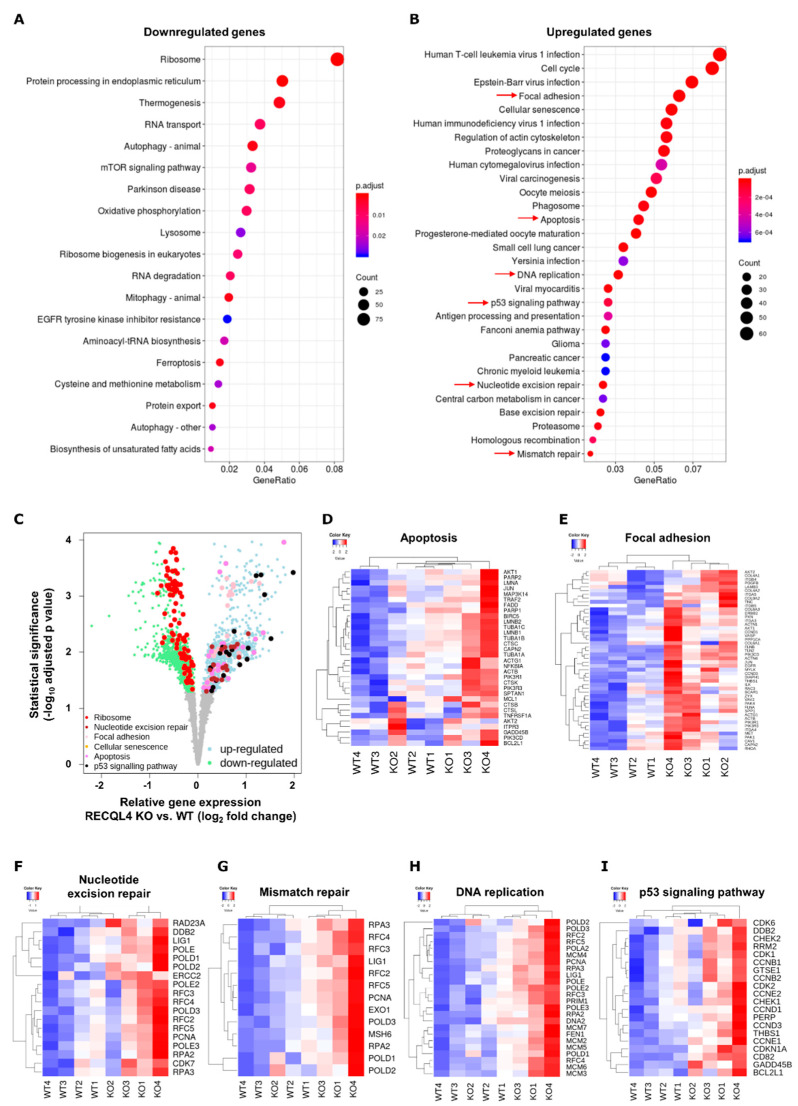
RNA-seq reveals gross transcriptomic changes in RECQL4-depleted glioma cells. Total RNA from WT and RECQL4 KO#2 glioma cells were subjected to RNA sequencing. KEGG analysis of DE genes (FDR corrected *p* < 0.05) shows genes/pathways downregulated (**A**) or upregulated (**B**) in RECQ4 KO cells when compared to WT cells. (**C**) Volcano plot shows down- and upregulated genes (log_2_ fold change < 0 and log_2_ fold change > 0, respectively, and FDR-correction *p* < 0.05) in RECQ4 KO cell lines relative to the WT. Genes from selected functional KEGG categories (*q*-value < 0.05) as marked by red arrows on panel (**B**) are marked in different colours, as indicated in the legend in bottom left of the panel (**C**). Selected KEGG pathways are also presented on z-score heatmaps: apoptosis (**D**) focal adhesion (**E**) nucleotide excision repair (**F**) mismatch repair (**G**) DNA replication (**H**) and p53 signalling pathway (**I**).

**Figure 4 cancers-12-02919-f004:**
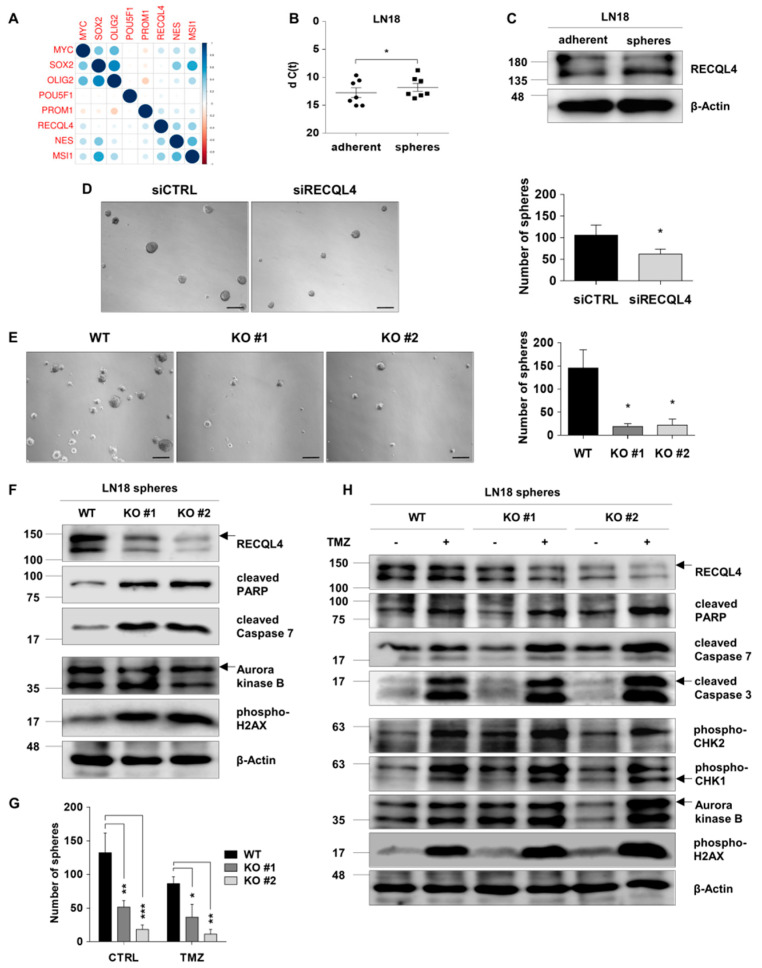
RECQL4 supports self-renewal of glioma stem cells and their resistance to temozolomide. (**A**) Pearson’s correlation between expression of *RECQL4* and selected stem cell markers in the TCGA dataset showed significant positive correlation of *RECQL4* and *MYC, NES, PROM1* and *MSI1* genes. (**B**) qPCR analysis of the *RECQL4* mRNA level in LN18 spheres and parental adherent cells. The expression was normalized to *GAPDH*; the results represent mean ± SEM (*n* = 7 different sphere cultures). Statistical significance was determined by two-tailed paired *t*-test. *p*-Values were considered as significant when * *p* < 0.05. (**C**) Representative immunoblot showing RECQL4 expression in LN18 spheres and adherent cells. β-Actin was used as a loading control. (**D)** Representative images and quantification of LN18 spheres (at day 7th) after siRNA-mediated RECQL4 knockdown compared to control siRNA transfected spheres. Scale bar represents 200 µm. The results are shown as numbers of spheres (≥100 µm in diameter were counted) and represent means ± SEM (*n* = 4). Statistical significance was determined by two-tailed paired t-test. *p*-Values were considered as significant when * *p* < 0.05. (**E**) Representative images and quantification of (at day 7th) WT and RECQL4 knockout LN18 spheres. (**F**) Representative immunoblots showing analysis of cell death, proliferation and cell cycle proteins in WT and RECQL4 KO LN18 spheres. (**G**) Quantification of spheres in WT and RECQL4 KO cultures exposed to 500 µM TMZ for 72 h. The results are shown as numbers of spheres and represent the means ± SEM (*n* = 3). Statistical significance was determined by RM two-way ANOVA, with followed by Tukey’s HSD post hoc test. P values were considered as significant when * *p* < 0.05, ** *p* < 0.01, *** *p* < 0.001. (**H**) Representative immunoblots show the levels of cell death, proliferation and cell cycle proteins in WT and RECQL4 KO LN18 spheres 72 h after TMZ treatment.

**Figure 5 cancers-12-02919-f005:**
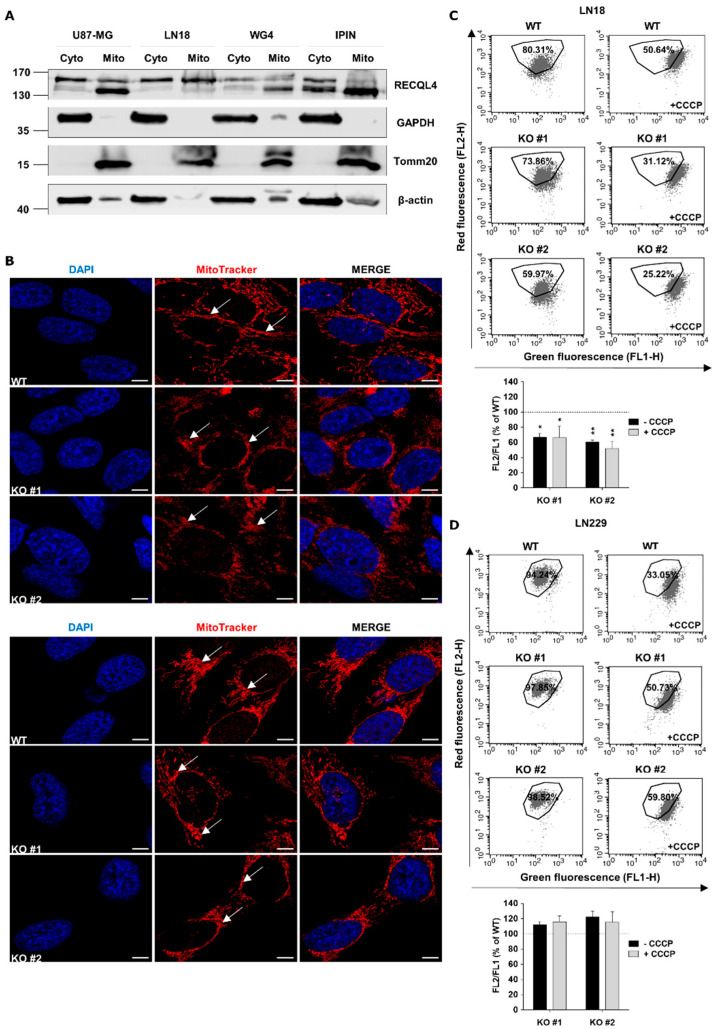
RECQL4 depletion affects morphology and functions of mitochondrial network in glioblastoma cells. (**A**) Representative immunoblots showing subcellular distribution of RECQL4 in cytosolic (Cyto) and mitochondrial (Mito) fractions of human U87-MG, LN18 and primary WG4, IPIN cultures of glioblastoma cells. Expression of GAPDH and Tomm20 was used as a loading control and markers for cytosolic and mitochondrial fractions, respectively. (**B**) Representative images of mitochondrial morphology and network organisation in LN18 (upper panel) and LN229 (bottom panel) cells analysed by confocal microscopy. WT and RECQL4 KO cells were stained for mitochondria (MitoTracker, red) and nuclei (DAPI, blue). Scale bar represents 5 µm. (**C**) Representative histograms and quantification of mitochondrial transmembrane potential changes in WT and KO LN18 cells. The cells were stained with JC-1 probe and analysed by FACS. CCCP was used as a positive control. The results are shown as red/ green fluorescence intensity ratio (relative FL-2/ FL-1) in RECQL4 KO cells normalized to WT cells, and represent means ± SEM (*n* = 3). Statistical significance was determined by RM two-way ANOVA, with followed by Tukey’s HSD post hoc test. P values were considered as significant when * *p* < 0.05, ** *p* < 0.01. (**D**) Representative histograms and quantification of mitochondrial transmembrane potential changes in WT and RECQL KO LN229 cells.

**Figure 6 cancers-12-02919-f006:**
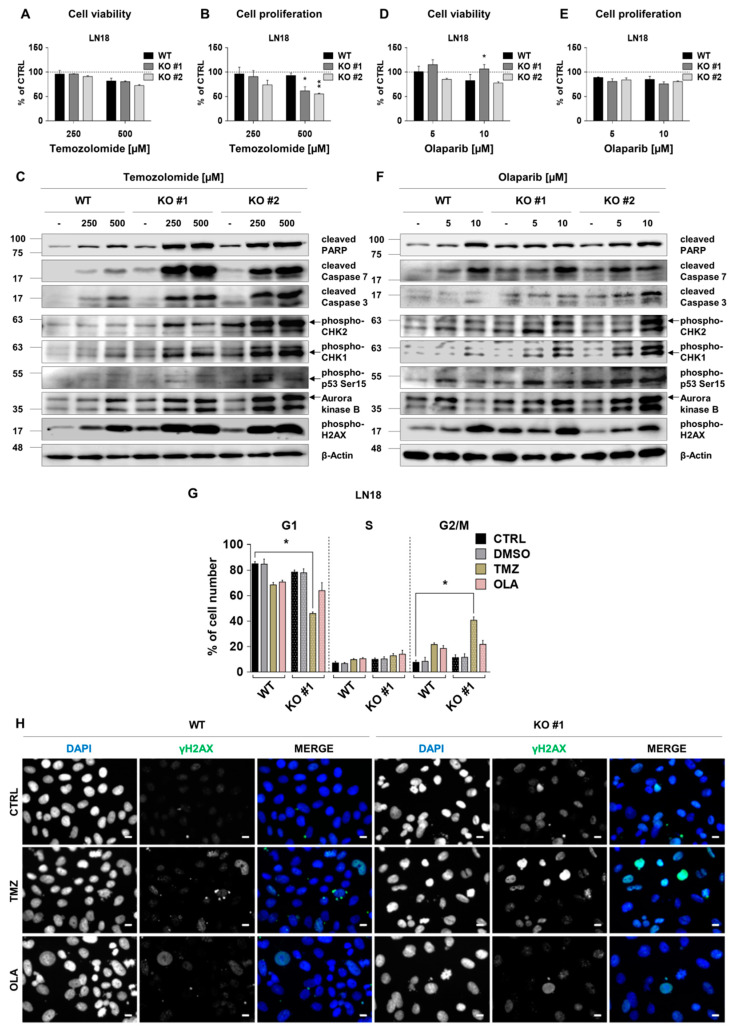
RECQL4 deficiency sensitises LN18 glioblastoma cells to temozolomide or olaparib. WT and RECQL4 KO LN18 cells were exposed to TMZ at concentrations of 250 and 500 µM for 72 h, and cell viability (**A**) and proliferation (**B**) were analysed. The results were normalized to untreated cells (CTRL) and represent means ± SEM (*n* = 4). Statistical significance was determined by RM two-way ANOVA, followed by Tukey’s HSD post hoc test. *p*-Values were considered as significant when * *p* < 0.05, ** *p* < 0.01. (**C**) Representative immunoblots showing the levels of cell death, proliferation and cell cycle proteins after TMZ treatment. β-Actin was used as a loading control. WT and RECQL4 KO LN18 were exposed to olaparib for 72 h and analysed for cell viability (**D**) and proliferation (**E**). The results were normalized to control cells (CTRL, with 0.05% DMSO as a solvent) and represent means ± SEM (*n* = 4). Statistical significance was calculated as above. (**F**) Representative immunoblots showing the levels of cell death, proliferation and cell cycle proteins after olaparib treatment in WT and RECQL4 KO cells. β-Actin was used as a loading control. (**G**) Cell cycle distribution analysis of WT and RECQL4 KO LN18 cells after TMZ (500 µM) or OLA (10 µM) treatment for 72 h. The results are shown as percentage of cell number in each phase of the cell cycle and represent means ± SEM (*n* = 3). Statistical significance was determined by Kruskal–Wallis one-way ANOVA followed by Dunnett’s post hoc test. *p*-Values were considered as significant when * *p* < 0.05., ** *p* < 0.01. (**H)** Representative images of immunofluorescent staining showing γH2AX foci and cell nuclei in WT and RECQL4 KO LN18 cells exposed to TMZ (500 µM) or OLA (10 µM) for 48 h. Scale bar represents 10 µm.

**Figure 7 cancers-12-02919-f007:**
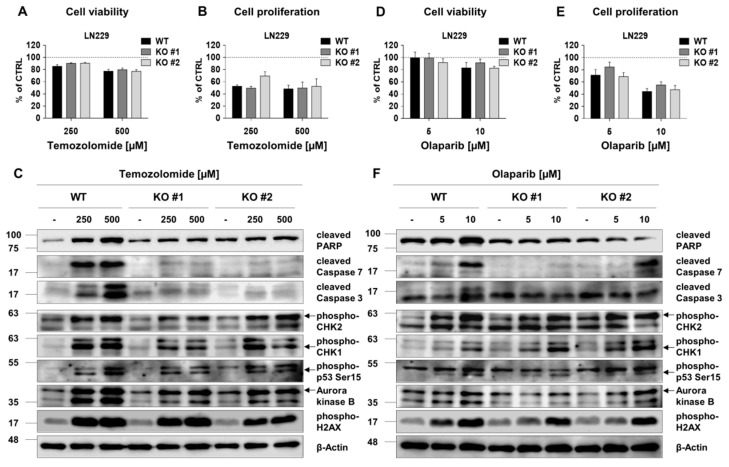
RECQL4 depletion does not sensitize LN229 glioma cells to temozolomide or olaparib. Analyses of viability and proliferation of WT and RECQL4-deficient LN229 cells exposed for 72 h to TMZ (250 and 500 µM) (**A,B**) or olaparib (5 and 10 µM) (**D,E**). The results were normalized to control cells (CTRL) and represent means ± SEM (*n* =4). Statistical significance was determined by RM two-way ANOVA, with followed by Tukey’s HSD post hoc test. Representative immunoblots show the levels of cell death, proliferation and cell cycle proteins after the treatments of WT and RECQL4 KO LN229 cells with TMZ (**C**) or olaparib (**F**). β-Actin was used as a loading control.

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
