# Peer review of "Aberrantly Expressed RECQL4 Helicase Supports Proliferation and Drug Resistance of Human Glioma Cells and Glioma Stem Cells"

_cancers, 2020, doi:10.3390/cancers12102919_

Round 1
Reviewer 1 Report
Aberrantly expressed RECQL4 helicase supports proliferation and drug resistance of human glioma cells and glioma stem cells
Sylwia K. Król , Agnieszka Kaczmarczyk , Kamil Wojnicki , Bartosz Wojtas , BartÅ‚omiej Gielniewski , WiesÅ‚awa Grajkowska , Katarzyna Kotulska-Jóźwiak , Cezary Szczylik , Ryszard Czepko , Mariusz Banach , Wojciech Kaspera , Wojciech Szopa , Andrzej Marchel , Tomasz Czernicki , Bozena Kaminska
Summary of the Key Findings of the Study:
The RECQL4 signaling pathway has a wide range of influences on normal cell physiology, and its deregulation plays a significant role in premature aging and predispose to cancer. In particular, RECQL4 signaling can promote rapid DNA repair reactivating thus cellular proliferation and cell migration, two central hallmarks of cancer. Therefore, the control and limitation of RECQL4 signaling remains essential in both breast and prostate cancer for prevention of neoplasia and tumor progression. Because Glioblastoma remains a devastating intracranial tumor resistant to chemo- and radiotherapy and because 50% of patient do not respond to the alkylating agent temozolomid, authors speculated that RECQL4 would provide a protective role.
General Critique of Work
Minor issues to be addressed:
Author names are spelled out and institutional affiliations are signified with footnotes. Likewise, corresponding authors are noted with an asterisk in the author list. Each author is well cited in the Author’s contributions part.
1/ Line 43, BLM and WRN genes remain not defined, “Bloom Syndrome RecQ Like Helicase, Werner Syndrome RecQ Like Helicase”
2/ Legend Figure 1, line 101, RECQL protein, authors mean RECQL4 protein or total RECQL proteins?
3/ Line 125: two “RQ4” KO cell lines, authors mean RECQL4 KO cell lines
Major issues to be addressed:
1/ Approval numbers from ethic committees to the design of study remains missing. Likewise, the study from the TCGA remains not referenced.
2/ Statistical analyses are different between the figure 1A and 1B, Dunnett’s test provides a multiple comparison to one control, while the Welch’s test compares at least 3 groups. Did authors have control the distribution of their data and the homogeneity of their variance?
3/ Likewise for the scoring method to evaluate RECQL4 immunostaining, the method remains missing and no statistical analysis were performed. How authors explain a cytoplasmic location of RECQL4. It will be useful to perform immunofluorescence on selected cell lines in parallel to the western blotting.
4/ How authors explain that RECQL4 is higher expressed in the conventional cell lines than in their own established glioma cells culture (WG4, IPIN). The difference would be explain by culture conditions (media) which may influence its expression or from the grade of the original patient as observed in the figure 1 A and B? Indeed, How authors explain that RECQL4 mRNA expression remains closed between the different cell lines and the primary cultures.
5/ Authors have elegantly developed CRISPR / Cas9 knockout cell lines, however, a background of RECQL4 expression remains visible. It’s a technical point, but authors have to address their sequencing results in supp. Data to estimate whether they obtained either a deletion, an insertion or a mix in the exon 3. Hence, authors would make sure to show the apparition of a STOP codon and not a disruption of the reading frame with a protein in downstream. Indeed, an alternative splicing would explain the RECQL4 background expression, as observed with the second bands closed to 130 kDa. Therefore, authors have established isogenic cell lines or a pool of puromycin resistant cells? KO#1 means clone 1? The method remains not clear? it will be useful to detail the method of clone selection to explain the residual RECQL4 expression.
6/ In LN18 cell line, both SiRNA and CRISPR cells show a weak decrease in cell proliferation and viability suggesting a weaker cell dependency for RECQL4 expression, i.e that RECQL4 remains not haploinsuffisant or dominant for LN18 viability. Hence, to conclude on the cell dependency, it will be interesting to reproduce SiRNA experiments on LN229 cells in manner to estimate whether RECQL4 deletion triggers similar results than obtained with CRISPR. Thus RECQL4 expression will be haploinsuffisant in this cell line. In such case, cell dependency for RECQL4 could be heterogeneous within glioma cell lines depending on the status of another oncogenic driver. Whatever, this crucial point have to be performed and discussed? For example, in the literature we found that the status of the MGMT seems different between the both cell lines.
7/ RNA Seq analysis remains confusing. KEGG representation is likely not crucial in this way, more in sup data. Inversely, Venn diagram to identify the number of genes shared by the different cell lines will be useful. Likewise, Heatmaps representation will be reduce to essential pathways. Authors investigate Apoptosis, Focal adhesion… and the next figure provides results from the TCGA concerning stem cells marker then apoptosis and DNA reparation. What is the basal status of stem cells marker obtained by RNA Seq? What is the Proneural, Mesenchymal, Classical and Neural signature, did RECQL4 KO cells orient cells in one of these signatures? The color legend for the Heatmaps remains not visual.
8/ Therefore, we observe that the KO2 cells remain closed to the WT3 and 4 in the following Heatmaps probably perhaps the non-KO cells from the clone selection have diluted KO cells. What is the method of clustering? Spearman, Pearson? For the Figure 4, authors provide a corrplot representation. What is the correlation coefficient? What is the method? The coefficient remains not really strong. It will be better to perform linear regression analysis from the CCLE and the TCGA portal. It is the same KO2 clones than used in heatmaps?
9/ RECQL4 deletion seems affect the both mitochondrial network and function, did authors have investigated this way though their RNA-Seq in manner to observe whether genes implicated in the mitochondrial dynamics remain unaltered as well as those brlonging to the respiratory chain? Which could be represent offtargets for the Cas9 in comparison to their parental cell lines. Therefor, immunofluorescence are not convaincing, it will be useful to have EM pictures.
10/ Likewise, the JC1 experiment remain the more sensitive method to appreciate the mitochondrial transmembrane potential, but usually, the JC1 labelling remains coupled with TOPRO staining to discriminate early and late apoptosis from living cells with high mitochondrial potential. Authors have to perform this experiment with a plasma membrane integrity dye and perform statistical analysis. Moreover, authors used CCCP to uncouple the respiratory chain and observed an increase effect in RECQL4 depleted cells. This phenomenon could also explain by the mitochondrial mass between parental and KO cells. Whatever, author have the response in their RNASeq analysis whether genes affecting mitochondrial functions and dynamism remain altered.
11/ Figure 6, concerning Cell viability and Proliferation experiments, the percentage of control remain calculated between KO cells and their parental cell line or the untreated KO cells? Likewise for the figure 7.
Blots of the total proteins for the phosphorylation remain missing. The decrease expression of MGMT is not show.
Line 349-350, the conclusion is different from the Figure 7 heading making difficult to understand the conclusion, or there is a confusion between the cell lines used?
Conclusion:
Authors have performed a good job with the elegant approach of the CRISPR/ Cas 9 method. However, this article is not suitable for publication in state. Authors have to improve their experimental approaches by taking in consideration the above remarks and perform supplemental experiments. In this way, authors have to review their conclusion, in particular for their observations toward mitochondrial function while the analysis of the geneset affecting mitochondrial function and structure remains missing following their RNASeq analysis. Likewise for the cell sensitivity to TMZ as previously mentioned.
Author Response
Responses to reviewer comments:
We thank the reviewers for their valuable comments and high appreciation of our manuscript. We found most comments justified and amended the text accordingly. We have addressed comments as follows:
Reviewer 1:
1/ Line 43, BLM and WRN genes remain not defined, “Bloom Syndrome RecQ Like Helicase, Werner Syndrome RecQ Like Helicase”
Ad. We followed the reviewer suggestions and provided full names.
2/ Legend Figure 1, line 101, RECQL protein, authors mean RECQL4 protein or total RECQL proteins?
Ad. It was omission, it has been correct to RECQL4, the main subject of our study.
3/ Line 125: two “RQ4” KO cell lines, authors mean RECQL4 KO cell lines
Ad. RQ4 is an abbreviation for RECQL4 siRNAs which were transfected (Fig.2A). We modified figure caotion(?) to explain this. In all other situation we meant RECQL4 KO cell lines. We unified this description to make it clear in the revised text.
Major issues to be addressed:
1/ Approval numbers from ethic committees to the design of study remains missing. Likewise, the study from the TCGA remains not referenced.
Ad. We provided a detailed information regarding Bioethics Committees and approval numbers in a separate document submitted to the Cancers office. We apologize for a lack of reference for TCGA. We introduced a reference for data downloaded from TCGA in the Methods section.
2/ Statistical analyses are different between the figure 1A and 1B, Dunnett’s test provides a multiple comparison to one control, while the Welch’s test compares at least 3 groups. Did authors have control the distribution of their data and the homogeneity of their variance?
Ad. In the TCGA dataset analysis (Fig 1A) the Welch’s ANOVA was used when gliomas of grades GII, GIII and GIV were compared. These data were not subjected to statistical testing, that is why a bracket and significance (***) is assigned to 3 right-hand side boxplots. In the TCGA dataset a number of normal brain samples is in our opinion too low in comparison to tumor specimens to make control-tumor statistical comparisons. Welch ANOVA is less sensitive to an unequal variance, which has a reason it was used in TCGA analyses, where a variance in GIV and GIII was greater than in GII.
For qPCR analysis presented in the figure 1B we used ANOVA + Dunnett’s post hoc test for NB vs. GI; NB vs. GII/III, NB vs. GIV comparison, as it was advised by our statistician. We did have more samples of normal brains in our own cohort. It showed that only glioblastoma significantly differs in terms of RECQL4 mRNA expression from a normal brain. In the presented qPCR analysis variance distribution was similar.
We apologize for not providing those important details. in the figure captions. We have extended the last sentence of the Fig1A caption: „Presented values are log2 of FPKM values. Statistical significance was determined by Welch’s analysis of variance (ANOVA) between GII, GIII and GIV groups.”
3/ Likewise for the scoring method to evaluate RECQL4 immunostaining, the method remains missing and no statistical analysis were performed. How authors explain a cytoplasmic location of RECQL4. It will be useful to perform immunofluorescence on selected cell lines in parallel to the western blotting.
Ad. The scoring was performed by two independent scientist based on positive staining of RECQL4 protein in nucleus, cytoplasm or both. The scoring was marked as 1 if staining was “present” or “0” – if absent. The resulting scores were analyzed using chi square test among the glioma grades. To visualize the RECQL4 location, we included representative images in figure 1D. We originally presented RECQL4 immunostaining, scoring and statistics description in the supplement, but following a reviewer comment, we moved it to the main text and extended a description.
Cytoplasmic staining of RECQL4 results from its presence in the mitochondria as this is the only Recq helicase found in mitoechondria. Zhenfen Chi et al. presented in the publication (doi: 10.1016/j.biocel.2012.07.016) that the NES-mediated RecQL4 export to the cytoplasm is essential for the maintenance of mitochondrial genome stability. Moreover, at the C-terminus of RECQL4 are presented two functional nuclear exporting signals (NESs; pNES2 and pNES3) which enable RECQL4 protein export from the nucleus. In figure 5A we presented fractionation of 4 cell lines showing presence the RECQL4 protein in cytoplasm (and in mitochondria) in established and primary cell lines.
4/ How authors explain that RECQL4 is higher expressed in the conventional cell lines than in their own established glioma cells culture (WG4, IPIN). The difference would be explain by culture conditions (media) which may influence its expression or from the grade of the original patient as observed in the figure 1 A and B? Indeed, How authors explain that RECQL4 mRNA expression remains closed between the different cell lines and the primary cultures.
Ad. Culture conditions (media) do not influence RECQL4 expression in conventional versus patient derived cultures as for an experiment those cells are cultured in the same media. Of course during establishment patient derived cultures are maintained in specific media and less subjected to exposure to serum which introduces most changes. It does not result from the grade of the original patient as all those cells lines are from glioblastomas. Based on our finding on sphere cultures, we may speculate that maintenance of established cell lines in media with serum depletes cancer stem cells with higher expression of RECQL4. Moreover, the primary cell lines are more heterogeneous, hence can have mix of cells of different RECQL4 expression. RECQL4 mRNA expression is similar in different cell lines and the primary cultures, but protein levels are different. Protein expression depends on the actual demand of the protein, and such discordances between mRNA and protein levels are frequent. The point is that both RECQL4 mRNA and protein levels are higher than in non-transformed astrocytes.
5/ Authors have elegantly developed CRISPR / Cas9 knockout cell lines, however, a background of RECQL4 expression remains visible. It’s a technical point, but authors have to address their sequencing results in supp. Data to estimate whether they obtained either a deletion, an insertion or a mix in the exon 3. Hence, authors would make sure to show the apparition of a STOP codon and not a disruption of the reading frame with a protein in downstream. Indeed, an alternative splicing would explain the RECQL4 background expression, as observed with the second bands closed to 130 kDa. Therefore, authors have established isogenic cell lines or a pool of puromycin resistant cells? KO#1 means clone 1? The method remains not clear? it will be useful to detail the method of clone selection to explain the residual RECQL4 expression.
Ad. In some experiments blots showed more complete RECQL4 knockdown, but more frequent was the level presented in the figure, so after discussion we decided to show this image despite potential criticisms. It could be a result of insufficient selection or as a reviewer has mentioned we can’t exclude that the RECQL4 background expression reflect alternative splicing. We expanded description of clone selection. We established the isogenic cell line(s) derived from puromycin resistant a single cell cultures Each RECQL4 KO cell line (LN18 as well LN229) is not the pool of cells, because after antibiotic selection the puro-resistant cells were plated as a single cell to form colonies. Then, after culturing, the clones derived from those single cells were validated for RECQL4 KO by Western Blotting and ultra-deep Next Generation Sequencing. Data from NGS showed deletions of short fragments in the RECQL4 gene. As KO #1 and KO #2 we marked two independently derived clones, derived from a single puro-resistant cell with the lowest expression of RECQL4 protein (as validated by Western blotting).
6/ In LN18 cell line, both SiRNA and CRISPR cells show a weak decrease in cell proliferation and viability suggesting a weaker cell dependency for RECQL4 expression, i.e that RECQL4 remains not haploinsuffisant or dominant for LN18 viability. Hence, to conclude on the cell dependency, it will be interesting to reproduce SiRNA experiments on LN229 cells in manner to estimate whether RECQL4 deletion triggers similar results than obtained with CRISPR. Thus RECQL4 expression will be haploinsuffisant in this cell line. In such case, cell dependency for RECQL4 could be heterogeneous within glioma cell lines depending on the status of another oncogenic driver. Whatever, this crucial point have to be performed and discussed? For example, in the literature we found that the status of the MGMT seems different between the both cell lines.
Ad. As RECQL4 depletion with siRNA in 2 cell lines did not affect significantly proliferation and viability, we switch to CRISPR. We believe if RECQL4 KO LN229 did show any effects on viability and proliferation it is likely that we would see effects of siRNAs. Indeed, LN18 and LN229 cell lines have different genetic and molecular background e.g. MGMT, TP53 status and therefore reflect heterogeneity and variability of observed in GBM patients. This may be a reason for different responses to CRISPR knockout of RECQL4 and drugs treatment. We agree that cell dependency for RECQL4 could be heterogeneous within glioma cell lines depending on the status of another oncogenic driver We addressed this point in the Discussion.
7/ RNA Seq analysis remains confusing. KEGG representation is likely not crucial in this way, more in sup data. Inversely, Venn diagram to identify the number of genes shared by the different cell lines will be useful. Likewise, Heatmaps representation will be reduce to essential pathways. Authors investigate Apoptosis, Focal adhesion… and the next figure provides results from the TCGA concerning stem cells marker then apoptosis and DNA reparation. What is the basal status of stem cells marker obtained by RNA Seq? What is the Proneural, Mesenchymal, Classical and Neural signature, did RECQL4 KO cells orient cells in one of these signatures? The color legend for the Heatmaps remains not visual.
Ad. The reviewer suggests an interesting analysis to find genes shared by the different cell lines, but we approach this issue from another angle. We focused on functions of common genes that change in RECQL4 depleted cells. That is why we did not present data in a Venn diagram to identify common genes shared by the different cell lines, To identify RECQL4 dependent transcriptional events, we focused on genes that are change in a similar manner in RECQL4 KO cells. RNAseq data from 2 cells lines were combined, the identity of a cell line was treated as a batch effect which was removed and we analyzed functional categories resulting from RECQL4 depletion. A heatmap scale is simple: red shows up and blue shows down regulated in respect to control. There is no data on RECQL4 functions in glioma cells and we found this aspect particularly interesting. We consider gene category heatmaps informative and showing good visualization. We did a broad analysis and found many categories, but due to space limitation we focused on those overrepresented and validated by biochemical and functional analyses (we showed data on cell cycle, cell death, stress responses after drugs).
We did not analyzed the basal status of stemness markers from RNA Seq data, as it is low in bulk cells. Even by qPCR their expression is low, unless cells as LN18 are cultured as spheres.
Proneural, Mesenchymal, Classical and Neural signatures are based on transcriptomic profiles from bulk tumors, and microenvironment and stromal cells contribute significantly to those signatures (in particular MES). We learnt that it is not advisable to apply signatures to cultured cells. We did try such approach to sphere cultures and some effects are visible, but I would not say it is reliable.
8/ Therefore, we observe that the KO2 cells remain closed to the WT3 and 4 in the following Heatmaps probably perhaps the non-KO cells from the clone selection have diluted KO cells. What is the method of clustering? Spearman, Pearson? For the Figure 4, authors provide a corrplot representation. What is the correlation coefficient? What is the method? The coefficient remains not really strong. It will be better to perform linear regression analysis from the CCLE and the TCGA portal. It is the same KO2 clones than used in heatmaps?
Ad. Pearson correlation performed on a large cohort of TCGA samples showed a positive significant correlation for MYC, NES, PROM1 and MSI1 correlation with RECQL4. Correlation coefficient for these 4 genes ranged from 0.183 to 0.335. Although these values seem low, we have to remember that correlation analysis was performed on 677 samples. Spearman’s correlation gave even better and more significant correlation coefficients, but it was discarded as it is not possible to compute exact p-values with ties in rank methods such as Spearman’s and we were afraid that such correlation results will be over-estimated.
We change description of the Fig4A caption to “Pearson’s correlation between expression of RECQL4 and selected stem cell markers in the TCGA dataset showed significant positive correlation of RECQL4 and MYC, NES, PROM1 and MSI1 genes.”
All diagrams, plots and heatmaps in the Figure 3 are obtained from RNAseq of the same set of samples:four replicates of WT and REQL4 KO#2. We introduced this information into the figure caption.
9/ RECQL4 deletion seems affect the both mitochondrial network and function, did authors have investigated this way though their RNA-Seq in manner to observe whether genes implicated in the mitochondrial dynamics remain unaltered as well as those brlonging to the respiratory chain? Which could be represent offtargets for the Cas9 in comparison to their parental cell lines. Therefore, immunofluorescence are not convincing, it will be useful to have EM pictures.
Ad. We experience difficulty in finding a good set of mitochondria-related genes as they are numerous. In this study, we wanted to find if mitochondrial network and functions change in RECQL4 depleted cells.
Regarding microscopy: we present what is standard in this type of studies: good quality, high resolution confocal microscopy pictures. Magnification of those images shows details at remarkable resolution. We are not experts in EM, so this is not available for us. The observed effects unlikely represent offtargets for the Cas9 in comparison to their parental cell lines because, we found those changes only in LN18 cells , not in LN229.
10/ Likewise, the JC1 experiment remain the more sensitive method to appreciate the mitochondrial transmembrane potential, but usually, the JC1 labelling remains coupled with TOPRO staining to discriminate early and late apoptosis from living cells with high mitochondrial potential. Authors have to perform this experiment with a plasma membrane integrity dye and perform statistical analysis. Moreover, authors used CCCP to uncouple the respiratory chain and observed an increase effect in RECQL4 depleted cells. This phenomenon could also explain by the mitochondrial mass between parental and KO cells. Whatever, author have the response in their RNASeq analysis whether genes affecting mitochondrial functions and dynamism remain altered.
Ad. We stained for mitochondria (MitoTracker, red) and nuclei (DAPI, blue) for confocal analysis. Mitochondrial transmembrane potential was measured with JC-1 in separate experiments and cells were under normal conditions, when there is no cell death. We actually measured mitochondrial mass with MitoTracker Green using Laser Scanning Cytometer, but as there was no statistically significant changes, we did not included those data.
11/ Figure 6, concerning Cell viability and Proliferation experiments, the percentage of control remain calculated between KO cells and their parental cell line or the untreated KO cells? Likewise for the figure 7.
Ad. In the figures 6 and 7. Control cells (CTRL=untreated) are cells with no treatment (for TMZ) or with 0.1% DMSO (solvent for OLA) in parental cells (WT) as well as in the cell lines with knockout of RECQL4 (KO # 1 and KO #2). The percentage of cells is calculated between CTRL (untreated) and treated cells – e.g. the percentage of WT cells after treatment (with TMZ or OLA) is calculated between CTRL WT (100%) and TMZ WT.
Blots of the total proteins for the phosphorylation remain missing.
Ad. This was mostly due to space constrain. For some of the protein we did detection of a total protein, but they levels in cells were similar (not shown). It is known that their levels do not change, but phosphorylation reflects their activation. We tried to make as many detections as possible on same membrane to make comparison more valid. Equal total loading was ensured by actin detection and Ponceau red staining of blots.
The decrease expression of MGMT is not show.
Ad. It is known from several studies that MGMT levels is reduced in LN229 (due to methylation of the MGMT gene promoter), while is high in LN18 cells non-methylated the MGMT gene promoter). We provided relevant references in the text. Moreover, we previously demonstrated that the MGMT gene promoter is unmethylated in LN18 cells by our group (Ciechomska, I. A., et al. doi: 10.3389/fphar.2018.01271).
High expression of MGMT at protein level in LN18 cells is shown in (doi.org/10.1111/j.1471-4159.2005.03583.x) and (https://doi.org/10.1111/j.1471-4159.2012.07781.x) and (Aasland, D., et al. https://doi.org/10.1111/jnc.14262). Methylation of MGMT in LN229 cells, and very low/lack expression at protein level have been demonstrated ( doi: 10.1158/0008-5472.CAN-10-1250; doi: 10.1371/journal.pone.0140131).
Line 349-350, the conclusion is different from the Figure 7 heading making difficult to understand the conclusion, or there is a confusion between the cell lines used?
Ad. There is no confusion. In Line 349-350 we refer to the role of RECQL4 in stemness of LN18 glioma cells (that were studied as LN229 cells do not grow as spheres). IN those cells RECQL4 depletion initiated cell death and sensitized to TMZ. In Fig. 7 we show data on LN229 cells in which RECQL4 knockout does not make a difference, as cells die after TMZ. This is an evidence to demonstrated that depending on genetic background of cells, the effects of RECQL4 depletion are more or less important.
Reviewer 2 Report
The manuscript by Krol et al describes how aberrant expression of RECQL4 support drug resistance in Glioblastoma (GBM). The studies focused on key and relevant topics in GBM by combining studies with the chemotherapeutics Temozolomide (TMZ) and Olaparib and as well as analysis of therapy resistant glioma stem cell populations. Overall, the studies have been well designed and the findings are interesting, but unfortunately much of it is observational and correlative. For example, take the RNA seq studies- none of the observations from this study were taken further for investigation in the manuscript. The paper also lacked mechanistic insight. For example- it is not clear how RECQL is involved in regulating any of the proteins described (direct-binding/transcriptionally).
Specific points:
- Figure 1C: Define high/low/medium expression levels
- Figure 2D and E: RECQL4 can still be detected via Western blots in CRISPR KO samples?
- Figure 3. It is not clear from the figure or the text what WT1-4 and KO1-4 samples are. Which cell line is it? Is that one cell line (WT) vs KO (repeated 4 times). From the methods it would seem this is two cell lines grouped?
- Figure 4G. RECQL4 KO does not appear reduce neurosphere formation in TMZ treated cells compared to control cells, (e.g. control KO#1 vs TMZ KO#1).
- Figure 5. Flow cytometry plots with JC-1 should be displayed as quadrants, and the results should be explained in a little more detail.
- Figure 6C. There is clearly DNA damage following TMZ treatment and this is exacerbated in the presence of RECQL4 KO. Is this due to increased accumulation of DNA damage or reduced DNA repair. Authors should explore by looking at y-H2AX foci induction and the rate of how these are resolved. A blot for MGMT should also be included.
- Explain why p53 is non-functional (line 253)? S15 phosphorylation would suggest activation of p53? Western blots for p21/MDM2 can be used to support the authors assertion.
Discussion
This section needs to actually discuss rather than summarise. Coupled with the lack of detailed interpretation of the results in the results section leaves many unanswered questions. More specifically the few “discussion points” included are very weak.
Line 337: “The augmented γH2AX level indicates an increase in DNA double-strand breaks (DSB)”: y-H2AX can indicate many other things! This is best determined via foci count- otherwise y-H2AX is simply a “general” measure of DNA damage.
Line 345: “Up-regulation of phospho-CHK1 and phospho-CHK2 in RECQL4 KO cells indicates growth arrest”, again CHK1/CHK2 Western blots are simple markers of DNA damage, and it remains unclear why cells are halting at G2/M cell cycle. Analysis of Wee1/CDC25C and G2/M specific cell cycle regulators CDK1/CyclinB should provide a potential mechanism of G2/M checkpoint activation.
Line 362: “We demonstrate that targeting of RECQL4 in glioma cells lacking functional TP53 could be a potential strategy”. Is it the lack of p53 or unmethylated MGMT (or both) that sensitives these cells to RECQL4 inhibition? This should be explored in additional cell lines.
Author Response
Reviewer 2:
The manuscript by Krol et al describes how aberrant expression of RECQL4 support drug resistance in Glioblastoma (GBM). The studies focused on key and relevant topics in GBM by combining studies with the chemotherapeutics Temozolomide (TMZ) and Olaparib and as well as analysis of therapy resistant glioma stem cell populations. Overall, the studies have been well designed and the findings are interesting, but unfortunately much of it is observational and correlative. For example, take the RNA seq studies- none of the observations from this study were taken further for investigation in the manuscript. The paper also lacked mechanistic insight. For example- it is not clear how RECQL is involved in regulating any of the proteins described (direct-binding/transcriptionally).
Ad. We disagree with a comment that most of our findings “ are interesting, but unfortunately much of it is observational and correlative”. Observation of GO categories overrepresented among differentially expressed after RECQL4 knockdown was driving us to validate some findings in glioma cells under basal conditions and treatment to therapeutics. It this study we were not able to provide answer if “RECQL is involved in regulating any of the proteins described (direct-binding/transcriptionally)”, but there is so little know about RECQL4 in gliomas that it is hard to study deeply each aspect. We focused on new roles of RECQL4 depletion in cancer stemness, mitochondria and drug responses. We continue these studies and will provide more answers in future. RECQL4 is not a transcription factor, so determining binding to specific genes is difficult and effects are likely indirect.
Specific points:
- Figure 1C: Define high/low/medium expression levels
Ad. We applied the TCGA definition of expression levels: Low: <0.5median’ Median: 0.5-2 median; High:>2median.
- Figure 2D and E: RECQL4 can still be detected via Western blots in CRISPR KO samples?
Ad. Indeed, there is residual RECQL4 levels CRISPR KO cells. In some experiments blots showed more complete RECQL4 knockdown, but more frequent it was at the level presented in the figure, so we decided to show this image as representative despite potential criticisms. It could be a result of insufficient selection or we can’t exclude that the RECQL4 background expression reflects alternative splicing. We expanded description of clone selection. We established the isogenic cell line(s) derived from puromycin resistant a single cell cultures Each RECQL4 KO cell line (LN18 as well LN229) is not the pool of cells, because after antibiotic selection the puro-resistant cells were plated as a single cell to form colonies. Then, after culturing, the clones derived from those single cells were validated for RECQL4 KO by Western Blotting and ultra-deep Next Generation Sequencing. Data from NGS showed deletions of short fragments in the RECQL4 gene.
- Figure 3. It is not clear from the figure or the text what WT1-4 and KO1-4 samples are. Which cell line is it? Is that one cell line (WT) vs KO (repeated 4 times). From the methods it would seem this is two cell lines grouped?
Ad. WT1-4 and KO1-4 samples are replicates (from 2 independent passages) of LN18 and LN229 WT and RECQL4 KO cells. We would like to determine the common effects of RECQL4 knockdown in two glioma cell lines, RNAseq data from two cell lines were grouped together and differentially expressed genes were identified . All diagrams, plots and heatmaps in the Figure 3 are obtained from RNAseq of the same set of samples. Heatmaps show expression in LN18 WT (x2), LN18 KO (x2), and LN229 WT (x2), LN229 KO (x2).
- Figure 4G. RECQL4 KO does not appear reduce neurosphere formation in TMZ treated cells compared to control cells, (e.g. control KO#1 vs TMZ KO#1).
Ad. It is correct. RECQL4 depletion so strongly affects sphere formation that addition of TMZ does not further reduce neurosphere formation in TMZ treated cells compared to control cells (e.g. control KO#1 vs TMZ KO).
- Figure 5. Flow cytometry plots with JC-1 should be displayed as quadrants, and the results should be explained in a little more detail.
Ad. We used the common way of presenting flow cytometry plots with JC-1, which is in our opinion very illustrative and shows clearly differences. Due to complexity of this picture we show only quadrants in which changes are observed. We added a description: “The % population of cells is given in the respective quadrants. Note the considerable reduction of % cells with high potential in RECQL4 KO cells. The right panel shows cells with mitochondria depolarized by treatment with CCCP).”
- Figure 6C. There is clearly DNA damage following TMZ treatment and this is exacerbated in the presence of RECQL4 KO. Is this due to increased accumulation of DNA damage or reduced DNA repair. Authors should explore by looking at y-H2AX foci induction and the rate of how these are resolved. A blot for MGMT should also be included.
Ad. Resolving a question if TMZ-induced DNA damage exacerbated in the presence of RECQL4 KO due to increased accumulation of DNA damage or reduced DNA repair, would require new experiments with kinetics to see how it is resolved (and many hours at confocal microscopy to provide quantification). The time allowed for a revision was too short to perform such experiments and provide answers.
We and others demonstrated reduced MGMT levels is in LN229, while it is high in LN18 cells non-methylated the MGMT gene promoter). We provided relevant references in the text. We previously demonstrated that the MGMT gene promoter is unmethylated in LN18 cells (Ciechomska, I. A., et al. doi: 10.3389/fphar.2018.01271). High expression of MGMT at protein level in LN18 cells is shown in (doi.org/10.1111/j.1471-4159.2005.03583.x) and (https://doi.org/10.1111/j.1471-4159.2012.07781.x) and (Aasland, D., et al. https://doi.org/10.1111/jnc.14262). Methylation of MGMT in LN229 cells, and very low/lack expression at protein level have been demonstrated ( doi: 10.1158/0008-5472.CAN-10-1250; doi: 10.1371/journal.pone.0140131).
- Explain why p53 is non-functional (line 253)? S15 phosphorylation would suggest activation of p53? Western blots for p21/MDM2 can be used to support the authors assertion.
Ad. The cell lines are from ATTC and their characteristics with a respect to TP53 is well known. We provided relevant references in the text. S15 phosphorylation of TP53 occurs but the gene is mutated and protein is not functional. At the moment we do not have extracts to perform Western blots for p21/MDM2 and repeating those experiments would take more time.
For example. Table 1: TP53 status and relative p53 and MGMT protein levels in the studied human GBM cell lines (https://doi.org/10.18632/oncotarget.11197)
Cell line |
TP53 status |
Relative p53 protein level |
Relative MGMT protein level |
||
Mean±SD |
p-value a |
Mean±SD |
p-value a |
||
T98/EV |
M237I |
1.0 |
- |
1.0 |
- |
T98/shRNA |
M237I |
0.7±0.49 |
<0.05 |
0.1±0.34 |
<0.05 |
U138 |
R273H |
1.0 |
n.s. |
0.6±0.13 |
<0.05 |
LN-18 |
C238S |
0.8±0.54 |
<0.05 |
1.2±0.28 |
<0.05 |
A172 |
R72P heterozygous SNP |
<0.1 |
<0.05 |
0 |
- |
U87MG |
Wild-type |
<0.1 |
<0.05 |
0 |
- |
U87/EV |
Wild-type |
<0.1 |
<0.05 |
0 |
- |
U87/MGMT |
Wild-type |
<0.1 |
<0.05 |
1.6±0.18 |
<0.05 |
Discussion
This section needs to actually discuss rather than summarise. Coupled with the lack of detailed interpretation of the results in the results section leaves many unanswered questions. More specifically the few “discussion points” included are very weak.
Ad. RECQL4 has not been studied in glioma before, so there is no data to compare with. We think that where strong data were available we provided interpretation, but we restrained from speculation. We agree that in some aspects we touched to surface instead of resolving problem, but due to time and space limitation we could not resolve each issue. WE continue those studies focusing now on specific details.
Line 337: “The augmented γH2AX level indicates an increase in DNA double-strand breaks (DSB)”: y-H2AX can indicate many other things! This is best determined via foci count- otherwise y-H2AX is simply a “general” measure of DNA damage.
Ad. It is generally accepted that the augmented γH2AX level indicates an increase in DNA double-strand breaks (DSB) among other things. This marker is frequently used to detect DBS after treatment of manipulation. Even without foci quantification we see the augmented γH2AX level in RECQL4 KO cells and it was corroborated by Western blot results.
Line 345: “Up-regulation of phospho-CHK1 and phospho-CHK2 in RECQL4 KO cells indicates growth arrest”, again CHK1/CHK2 Western blots are simple markers of DNA damage, and it remains unclear why cells are halting at G2/M cell cycle. Analysis of Wee1/CDC25C and G2/M specific cell cycle regulators CDK1/CyclinB should provide a potential mechanism of G2/M checkpoint activation.
Ad. During G2 many proteins collaborate to activate Chk1, an effector protein kinase that ensures the mitotic cyclin-dependent kinase remains in an inactive state. This allows time for completion of DNA repair before commitment to mitosis. That is why we check on phosphorylation status of CHK1/CHK2.Their increased levels suggest that DNA repair may not be completed resulting in growth arrest.
We did Western blots for Cyclins B1 i D1, but antibodies did not work properly and reliably in our hands and we did not included those results into this study. At the moment we can’t say about potential mechanism of G2/M checkpoint activation.
Line 362: “We demonstrate that targeting of RECQL4 in glioma cells lacking functional TP53 could be a potential strategy”. Is it the lack of p53 or unmethylated MGMT (or both) that sensitives these cells to RECQL4 inhibition? This should be explored in additional cell lines.
Ad. At the moment we can’t say if the lack of p53 or unmethylated MGMT are important, but as methylation of MGMT is generally tested before TMZ treatment, so it could be valid indication if status of RECQL4 is known. We publish at the Biorx a manuscript showing newly detected deleterious RECQL4 mutations in GBM (doi: https://doi.org/10.1101/2020.06.17.158477) and we are working on this issue as RECQL4 KO cells with reintroduced a mutated version seem to be more sensitive to TMZ. It is known that TMZ works in some patients with unmethylated MGMT, so it is tempting to suggests that it is a status of TP53 which matters. But without additional data, we do not dare to speculate.
Round 2
Reviewer 1 Report
Authors have taken the different remarks into consideration and have improved their manuscript.
These works can be published in Cancers.
Author Response
Authors have taken the different remarks into consideration and have improved their manuscript.
Ad. We thank a referee for accepting our answers and considering a revised version as an improved manuscript.
Reviewer 2 Report
In the revised version the author address and clarify some points but none of the suggested experiments have been conducted which would have strengthened the manuscript and necessary to provide a greater mechanistic understanding. The authors put this down to time constrain afforded by the editors but all of the experiments could easily be done in 1-2 months.
Author Response
Indeed normally it would take 1-2 months, but under current conditions it more complicated, as due to Covid-19 pandemics our institute work in shift, delivery of reagents is delayed. All these factors would further delay a revision. We are confident that the presented data support our claims and provide mechanistic insights. The experiments proposed by a referee 2 such as quantification of kinetics of γH2AX accumulation and resolution would require hours of work on confocal microscopy, which is simply not possible in a short time. Recently, our confocal facility was closed for two weeks due to quarantine.
Besides, I not fully agree that the proposed experiments are necessary. These experiments would While I agree, that they would strengthened the manuscript and provide a greater mechanistic understanding, they are designed not to verify or validate our results but to expand a scope of the manuscript. For example adding an experiment with siRNA mediated depletion on LN229 cells, would not provide an novel or important information as most experiments were done on CRISPR KO cells (Figs.2-7). We have shown data on effects of siRNA mediated depletion on two cells lines as controls to demonstrate that effects are similar as in CRISPR KO cells.
Round 3
Reviewer 2 Report
Clearly no additional work has been conducted in the revised version. And the author don’t intend to. Unfortunately their reason re pandemic challenges is a real possibility. I think all labs are experiencing this to one extent or another.
Editorial board should make a final decision.